# Concentrate Apple Juice Industry: Aroma and Pomace Valuation as Food Ingredients

Elisabete Coelho [1],*, Mariana Pinto [1], Rita Bastos [1], Marco Cruz [1], Cláudia Nunes [1,2], Sílvia M. Rocha [1] and Manuel A. Coimbra [1]

1   LAQV-REQUIMTE, Department of Chemistry, University of Aveiro, 3810-193 Aveiro, Portugal;
    a39045@ua.pt (M.P.); ritabastos@ua.pt (R.B.); mcruz@ua.pt (M.C.); claudianunes@ua.pt (C.N.);
    smrocha@ua.pt (S.M.R.); mac@ua.pt (M.A.C.)
2   CICECO—Aveiro Institute of Materials, Department of Chemistry, University of Aveiro,
    3810-193 Aveiro, Portugal
*   Correspondence: ecoelho@ua.pt; Tel.: +351-234-370706

**Abstract:** Apple concentrate juice industry generates a flavored coproduct (apple aroma) recovered in the evaporation process, which is poorly valuated due to the lack of chemical characterization and standardization. In this study, industry apple aroma was characterized, allowing for the identification of 37 compounds, the majority esters (20), alcohols (7), and aldehydes (4). The storage temperature did not affect its volatile composition. Five key compounds were selected and monitored for 10 months of storage, and also compared with other three productions of another season allowing for observation of the same Aroma Index. Apple pomace was also used to produce a hydrodistillate. Contrary to the apple aroma, apple pomace hydrodistillate was unpleasant, reflected in a different volatile composition. Although no additional aroma fraction could be obtained from this wet byproduct, when dried, apple pomace presented 15 volatile compounds with toasted, caramel, sweet, and green notes. The infusions prepared with the dried apple pomace exhibited 25 volatile compounds with a very pleasant (fruity, apple-like, citrus, and spicy notes) and intense aroma. The addition of sugar changed the volatile profile, providing a less intense flavor, with almond, caramel, and sweet notes. These results show that apple aroma and pomace are high-quality flavoring agents with high potential of valuation as food ingredients.

**Keywords:** volatile compounds; condensed aroma; Aroma Index; odor active value; byproduct; coproduct; infusion; decoction; sugar addition; HS–SPME/GC–MS



## 1. Introduction

Concentrate juice industry generates several coproducts and byproducts. The apple aroma, condensed volatiles obtained from the concentration process, is considered a coproduct when its composition is standardized and rich in the volatile compounds with apple scent. The apple pomace obtained from pressing and retentate obtained from ultrafiltration process are usually considered as byproducts [1].

During the concentration of the fruit juices, the volatile compounds are co-evaporated with the water. The evaporation rate of the aroma compounds depends on its volatility, a process that is completely distinct from an apple, plum, apricot, or strawberry juice, decreasing its aroma volatility in this sequence. Apple aroma can be completely lost at an evaporation degree of 15%, i.e., 15 L of volume evaporated in 100 L of juice results in a complete loss of apple aroma. Due to its high volatility, it is important to minimize the aroma losses during the beverage processing [2]. The technologies used for aroma recovery have the objective to minimize the losses of aroma by producing an aroma concentrate which can be used to reconstitute the final products, e.g., apple juices [3] or ciders produced from concentrated juice. The term aroma concentrate means a process stream in which the aroma has been separated and concentrated 100–200 times, or even

higher, with respect to the inflow of raw material. The production of aroma concentrates can be achieved by different technologies, the most common are vapor–liquid separations based on distillation/evaporation or on partial condensation and gas injection-based separations. Other technologies less usual at industrial level are adsorption, supercritical fluid extraction, and pervaporation [2,4]. The pervaporation uses a dilute apple essence, the aroma concentrates from vapor–liquid separations, to produce concentrated flavor products. The apple aroma can be used in both forms as more dilute or as a concentrated flavor product. Especially in the food and beverage industry, concentrated flavors are in demand due to prolonged shelf life, reduced packaging volume, and lower distribution and storage costs. This demand has triggered its industrial application. However, there are difficulties in the pervaporation concentration process to maintain the original flavor. The enrichment of the compounds differs from their chemical families, being the alcohols that had the lowest enrichment factors. The current concentration pervaporation methods are unable to keep exactly the original flavor after concentration; hence, flavor adjustments are necessary after concentration processes [5]. The obtention of a concentrated apple aroma with the presence of key flavor compounds is essential to restore the original fresh apple flavors to the final product. The vapor–liquid separations continue as the industrial technology most used, possibly to keep the original flavor of the aroma recovered, and its standardization over the processing.

Several strategies for the valuation of apple pomace have been proposed. Apple pomace has been used directly for animal feed, as a source of food ingredients [6], for pectin recovery [1], for extraction of phytochemicals [1,7], bioproduction of high value-added products, such as enzymes, organic acids, and biofuels [8], and the development of new materials as biocomposites and scaffold for tissue engineering [9]. However, for the retentate fewer applications have been proposed, such as bioproduction of citric acid [8,10], insect diets [8], and as a nutritive medium for production of apple cider and feed formulations for racing pigeons [11]. Despite these different strategies, the perishable nature of these byproducts, due to their high water content and huge amounts of nutrients, makes its valuation difficult; to be profitable, it needs to be integrated into the plant production in order to use the byproducts immediately. Apple pomace valuation is sometimes not profitable because of incoming costs for its stabilization and then use in wetness processes. Apple peels, a component of apple pomace, have been dried to be used in herbal tea production; the best conditions were achieved in terms of both drying time and color and nutrients [12]. Neither the volatile compounds of the dried product were assessed nor the evaluation of the aroma compounds present in the infusion.

In the present study, the apple aroma obtained during the evaporation step from the concentrate juice process was characterized and its storage stability was evaluated. The Aroma Index of industrial apple aroma using samples from different season productions was compared. The apple pomace hydrodistillate was evaluated concerning the valorization of volatile compounds present and its contribution to the apple aroma. The apple pomace infusions were proposed as a direct application to apple pomace byproduct. The volatile compositions of the brews prepared with and without sugar were also accessed to mimc the consumer preferences. To fulfil this objective, three different strategies of valorization were evaluated: (a) the quality assessment of the apple aroma produced during juice evaporation step from different production seasons and its definition as a coproduct, (b) the recovery of an additional apple aroma by hydrodistillation from the wet apple pomace, and (c) the stabilization of apple pomace as a dried product for infusions preparation.

## 2. Materials and Methods

### 2.1. Samples

The apple aroma samples were recovered from an industrial apple juice concentrate process at Indumape S.A. (Pombal, Portugal), using mainly Royal Gala apple variety, during January 2014 (apples from controlled atmosphere harvested in September/October 2013), September 2014, October 2014, and November 2014 (apples processed after harvest-

ing or stored under refrigerated conditions) and stored in 1.5 L plastic bottles at 4 °C and room temperature for 10 months. The industrial aroma samples were obtained during the concentration of the raw juice with a column heated at 90–120 °C, until the juice attained 20 °Brix.

Apple pomace was collected during January 2014 after apple pressing in the Indumape SA plant. The wet apple pomace was used directly for the hydrodistillation process and also stored in plastic bags in a freezer (−20 °C). The dried pomace was obtained after defrosting (at 4 °C, overnight) apple pomace by hot-air drying at 80 °C in an oven with ventilation for 4 h, giving a final moisture of 5%. Dried pomace was stored (1 week) in sealed plastic bags until analysis and infusion preparation.

### 2.2. Volatile Compounds Analysis by Headspace–Solid Phase Microextraction Gas Chromatography Combined with Quadrupole Mass Spectrometry (HS–SPME/GC–qMS)

The aroma recovered from the apple juice concentrate industry (the condensed water obtained during the juice evaporation process) was analyzed after dilution in water (1:10). Diluted sample (200 μL) was added to 2.8 mL of water into an 8 mL glass vial with 0.6 g NaCl. The first fraction of the hydrodistillate recovered from apple pomace (24 mL of each replicate) was also analyzed using the same SPME conditions without dilution. The samples were placed in a thermostat-controlled bath at 40 °C with stirring at 3800 rpm, DVB/CAR/PDMS (divinylbenzene/carboxen/poly(dimethylsiloxane)—50/30 μm) Stable-Flex™ fibers, with 1 cm length inserted into the sample vial headspace and the compounds were extracted for 30 min. The volatile compounds were injected in the GC injection port at 250 °C in splitless mode, separated, and identified in a gas chromatograph (Agilent Technologies 6890 N Network) equipped with a 30 m × 0.32 mm i.d., 0.25 μm film thickness polyethylene glycol-2-nitroterephthalate (DB-FFAP—Free Fatty Acid Phase) fused silica capillary column, connected to an Agilent 5973 mass selective detector. The GC was programmed from 40 to 80 °C at 2 °C/min, followed by 80 to 220 °C (2 min) at 10 °C/min. The transfer line was heated at 255 °C and the He carrier gas had a flow of 1.7 mL/min. The mass spectrometer was operated in the electron impact mode at 70 eV scanning the range 30–300 *m/z*. Identification of volatile compounds was achieved comparing the GC retention times and mass spectra with those of the pure standard compounds, when available. All mass spectra were also compared with the data system library (Wiley 275). The Relative Aroma Index was calculated by dividing the GC peak area $\times\ 10^{-7}$, mean of three replicates, divided by its odor threshold in mg/L (found in literature [13–17]).

For quantification purposes the key odorant compounds were analyzed by HS–SPME/GC–flame ionization detector (FID). The quality assessment of apple aroma was performed by quantification of the key aroma compounds present in apple aroma from the four seasons of production and storage. The quantification of the key odorant compounds was achieved by using the external calibration curve for each compound. The calibration curves were constructed with five concentration levels with two replicates of each standard: ethyl acetate ($y = 1 \times 10^{10}x - 4228.6$; $1.9$–$7.5 \times 10^{-5}$ mg/L), ethyl 2-methylbutanoate ($y = 3 \times 10^{8}x - 42,802$; $7.1$–$21 \times 10^{-4}$ mg/L), hexanal ($y = 7 \times 10^{7}x - 7549.3$; $1.7$–$20 \times 10^{-3}$ mg/L), butanol ($Y = 465,366x + 233,362$; $0.26$–$2.6$ mg/L), and *trans*-2-hexenal ($Y = 2 \times 10^{7}x - 738,574$; $5.2$–$21 \times 10^{-2}$ mg/L). A Perkin Elmer Clarus 400 gas chromatograph was used with a splitless injector. The injection port and detector were set to 250 °C. The GC oven temperature starts with a 5 min isothermal at 35 °C and then raised at 2 °C/min to 80 °C, followed by 80 to 220 °C (1 min) at 20 °C/min. The Aroma Index was calculated dividing the concentration, mean of two independent samples of at least two and a maximum of five replicates, by its odor threshold, both in μg/L.

### 2.3. Volatile Compounds Extraction by Simultaneous Distillation Extraction (SDE) and Analysis by GC–qMS

The aroma was obtained during the juice evaporation process from the apple juice concentrate industry (25 mL), diluted with distilled water (475 mL), and the addition of 188 μg of pentanoic acid (>99%, Aldrich), using 100 μL of a 2 μL/mL solution as the

internal standard subjected to simultaneous distillation extraction (SDE, 2 h) with 70 mL of pentane in a modified Likens–Nickerson apparatus. The extracts obtained were dried over anhydrous sodium sulfate and concentrated by distillation using a Vigreux column under low pressure at room temperature and a trap with liquid nitrogen on top [3,18]. The concentrated extracts were stored in a glass screw-top vial at −20 °C until being analyzed by GC–qMS. The detector started to operate after 8 min of injection (solvent delay) using the same GC temperature and MS acquisition conditions noted in Section 2.2. Estimated concentrations were made by GC peak area comparisons of the SDE extract components with the area of internal standard (pentanoic acid). The Aroma Index was calculated dividing the concentration by its odor threshold, both in µg/L.

### 2.4. Volatile Compounds Analysis of Dried Apple Pomace and Infusions by HS–SPME/GC–qMS

The dried apple pomace (12 g) was inserted in a 120 mL vial (1/3 of the vial volume) and placed in a thermostat-controlled bath at 60 °C. The SPME fiber (DVB/CAR/PDMS) was inserted into the sample vial headspace during 20 min for compound extraction. The infusions were prepared directly in the vial using a ratio of 2 g/100 mL of distilled water with 15 min agitation [19], using a total of 40 mL of boiling water, maintaining the same $1/\beta$ ratio of dried apple pomace. The $1/\beta$ ratio (ratio of the volume of the liquid/solid phase to the headspace volume) is an important variable in SPME analysis that should be maintained the same in both experiments. Headspace volume is a critical factor for determining the results in three-phase systems—liquid/solid sample–headspace–fiber coating [20]. Afterward, the vial was placed in a thermostat-controlled bath at 60 °C and the DVB/CAR/PDMS SPME fiber was inserted for 20 min [21]. In the preparation of the infusion with sugar, 2 g of sucrose was added to the vial before water addition. The sugar amount was defined taking in account that two packets (6 g per packet) is used to sweeten 1 cup of tea (240 mL). All experiments were done in triplicate. The volatile compounds were thermically desorbed in the injection port, separated, and identified by an Agilent Technologies 6890 N Network gas chromatograph using the same GC–qMS conditions referred to in Section 2.2.

### 2.5. Data Processing

Statistical analyses were performed to evaluate the effect of the storage temperature on the volatile compounds of apple aroma obtained upon the industrial juice evaporation step, considered statistically significant when $p < 0.05$. Analysis was performed by a multiple t-test (Tukey's HSD), using the GraphPad Prism version 6.01 for windows (trial version, GraphPad Software, San Diego CA, USA). A heatmap visualization of the data corresponding to the GC peak area of volatile compounds, organized by chemical families (apple aroma obtained upon the industrial juice evaporation step and hydrodistillate obtained from wet apple pomace; dried apple pomace and apple pomace infusions) was performed after applying the maximum normalization function of each GC peak area using the Unscrambler®X (30-day trial version—CAMO Software AS, Oslo, Norway).

## 3. Results and Discussion

### 3.1. Volatile Composition of Industrial Apple Aroma

The aqueous aroma solution recovered from the evaporation/distillation step of apple juice concentrate industry is usually stored at room temperature before expedition for periods that could last 1 month or even more. Due to the high volatility of apple aroma [2] combined with the low water solubility, aroma losses may result during storage. In this study, the volatile composition of industrial apple aroma stored after 1 month at room temperature and refrigerated at 4 °C was screened by HS–SPME/GC–qMS. The volatile profile of the industrial apple aroma is presented in Table 1.

**Table 1.** Volatile composition of apple aroma stored at 4 °C and room temperature, aroma descriptors, and odor threshold reported in literature.

| Compound | Retention Time (min) | Chromatographic Peak Area ($10^{-7}$) | | Mean (n = 6) | SD | Aroma Descriptor [1] | Odor Threshold (ppb) [1] |
| | | Storage Temperature | | | | | |
| | | 4 °C (n = 3) | 20 °C (n = 3) | | | | |
| *Acids* | | | | | | | |
| Nonanoic acid | 37.7 | 0.01 | 0.03 | 0.02 | 0.01 | Green, Fat | 3000 |
| *Alcohols* | | | | | | | |
| Ethanol | 3.7 | 0.40 | 0.49 | 0.44 | 0.07 | Slight, Sweet | 100,000 |
| 1-Butanol | 10.5 | 0.43 | 0.66 | 0.55 | 0.16 | Sweet, Malty, Solvent-like | 500 |
| 2-Methyl-1-butanol | 13.8 | 0.28 | 0.47 | 0.38 | 0.11 | Highly diluted–pleasant, Malty | 300 |
| 1-Hexanol | 22.7 | 9.50 * | 11.74 * | 10.62 | 1.38 | Herbaceous, Green, Fruity, Slightly fatty odor | 500 |
| *Trans*-2-hexenol | 25.2 | 1.14 | 1.31 | 1.23 | 0.10 | Green, Leaf, Walnut | 400 |
| 1-Octanol | 30.1 | 0.08 | 0.12 | 0.10 | 0.03 | Chemical, Metal, Burnt | 110 |
| 1-Nonanol | 31.9 | 0.03 | 0.03 | 0.03 | 0.01 | Fat, Green | 50 |
| *Aldehydes* | | | | | | | |
| Acetaldehyde | 1.9 | 0.08 | 0.11 | 0.09 | 0.03 | Green, Sweat, Fruity, Pungent | 17 |
| Hexanal | 7.2 | 2.12 | 2.78 | 2.45 | 0.55 | Green, Grass | 11 |
| *Trans*-2-hexenal | 13.9 | 4.73 | 5.52 | 5.12 | 0.55 | Green, Apple | 17 |
| Benzaldehyde | 29.4 | 0.15 | 0.15 | 0.15 | 0.02 | Bitter almond, Green almond, Burnt sugar | 350 |
| *Esters* | | | | | | | |
| Ethyl acetate | 2.9 | 2.06 | 2.15 | 2.11 | 0.30 | Ethereal–Fruity, Pleasant | 13,500 |
| Ethyl propionate | 3.8 | 0.13 | 0.22 | 0.18 | 0.09 | Sweet, Apple | 10 |
| Propyl acetate | 4.1 | 0.15 | 0.22 | 0.18 | 0.05 | Pear, Raspberry | 2000 |
| Methyl butanoate | 4.3 | | 0.08 | 0.08 | 0.00 | Apple, Fruity, Sweet | 59 |
| Ethyl butanoate | 5.6 | 1.91 | 2.42 | 2.17 | 0.37 | Fruity, Apple | 1 |
| Ethyl 2-methylbutanoate | 6.1 | 2.44 | 3.15 | 2.80 | 0.58 | Sweet, Fruity, Strawberry, Blackberry, Green apple | 0.006 |
| Butyl acetate | 6.9 | 7.24 * | 8.47 * | 7.85 | 1.12 | Red apple aroma | 66 |
| 2-Methylbutyl acetate | 8.7 | 4.84 * | 5.89 * | 5.37 | 0.86 | Fruity | 11 |
| Butyl propanoate | 9.5 | | 0.35 | 0.35 | 0.09 | Sweet, Fruity | 25 |
| Pentyl acetate | 11.3 | 0.25 | 0.34 | 0.30 | 0.07 | Fruity, Apple | 43 |
| Butyl 2-methylbutanoate | 14.3 | 0.70 | 0.87 | 0.78 | 0.21 | Fruity, Apple | 17 |
| Ethyl hexanoate | 14.5 | 0.58 | 0.75 | 0.66 | 0.23 | Fruity, Apple peel | 1 |
| Hexyl acetate | 17.1 | 10.29 * | 11.84 * | 11.07 | 1.06 | Herbaceous, Fruity | 2 |
| (E)-2-Hexenyl acetate | 20.3 | 0.61 | 0.69 | 0.65 | 0.05 | Green, Fruity, Sweet | 320 |
| Butyl hexanoate | 25.4 | 0.10 | 0.11 | 0.11 | 0.01 | Green apple, Fruity | 700 |
| Hexyl butanoate | 25.6 | 0.24 | 0.28 | 0.26 | 0.02 | Apple, Fruity | 6400 |
| Hexyl 2-methylbutanoate | 26.2 | 0.92 | 0.82 | 0.87 | 0.08 | Strawberry, Fruity, Fresh green | 6 |
| Benzyl Acetate | 33.0 | 0.01 | 0.01 | 0.01 | 0.00 | Menthol, Woody, Honey, Rain, Pear | 364 |
| 2-phenylethyl acetate | 34.1 | 0.01 | 0.01 | 0.01 | 0.00 | Sweet, Pipe tobacco, Roses, Honey | 480 |
| (E,Z)-2,4-Ethyl decadienoate | 35.2 | 0.04 | | 0.04 | 0.01 | Pear | 100 |
| *Phenols* | | | | | | | |
| Estragole | 32.0 | 0.50 | 0.41 | 0.46 | 0.08 | Licorice, Anise | 16 |
| *Trans*-anethole | 33.0 | 0.03 | 0.02 | 0.02 | 0.01 | Anise-like | 73 |
| Phenol | 36.2 | | 0.002 | 0.002 | 0.001 | Phenol | 5900 |
| Methyleugenol | 36.3 | 0.01 | | 0.01 | 0.00 | Clove, Spice | 820 |
| *Terpenoids* | | | | | | | |
| Geranylacetone | 34.5 | 0.05 | 0.05 | 0.05 | 0.01 | Fresh-floral, Light, Sweet-rosy | 60 |

[1] The odor descriptor and thresholds in water reported in literature [13–17]. * Values significantly different between the two storage conditions, $p < 0.05$.

The volatile composition of industrial apple aroma showed the presence of 37 compounds, the majority esters (20), followed by alcohols (7), aldehydes (4), phenols (4), 1 terpenoid, and 1 acid (Table 1). The prevalence of esters and alcohols is in accordance with the volatile composition of apple juices and aroma [3,13,22]. Slight differences were observed in the volatile composition of the industrial apple aroma stored at room temperature in comparison with the refrigerated conditions. There were only two compounds ((E,Z)-2,4-ethyl decadienoate and methyleugenol) that were not detected at room tem-

perature storage. However, these compounds were present in the sample stored at 4 °C with a GC peak area <0.05 × 10$^7$. Otherwise, two other compounds, the esters methyl butanoate and butyl propanoate, were not detected in the sample stored at refrigerated conditions. The chromatographic area of the majority of the volatile compounds presented in the apple aroma stored at different temperatures were not significantly different, $p < 0.05$. Only four compounds had significantly higher areas at room temperature: 1-hexanol, butyl acetate, 2-methylbutyl acetate, and hexyl acetate. Therefore, these results highlight that temperature storage conditions seem not to affect the volatile profile of the industrial apple aroma. Thus, it is more economically favorable for storage at room temperature in closed containers.

Overall, the major compounds present in apple aroma were hexyl acetate and hexanol, both with herbaceous and fruity descriptors, but with very different odor thresholds, 2 and 500 ppb, respectively (Table 1). In order to relate the volatile profile obtained with the concentration of each compound present, extractions of volatile compounds using SDE and quantification by GC–qMS using pentanoic acid as an internal standard were performed. A total of 17 volatile compounds were quantified (Table 2). The compound present in major concentration was hexanol (1426 mg/L), followed by butanol (549 mg/L), *trans*-2-hexenol (280 mg/L), and *trans* 2-hexenal (112 mg/L), in accordance with the apple aroma volatile concentration obtained by several industrial samples from different origin countries [3]. Undesirable compounds such as 3-methyl-1-butanol and 3-methylbutyl acetate, associated with fermentative metabolites produced under industrial apple fruit juice processing conditions [3,23], were not detected in the present study. In addition, furfural, a compound associated with thermal processing [24], was not present in the industrial apple aroma under study.

**Table 2.** SDE/GC–qMS volatile compound quantification of apple aroma stored at room temperature, aroma descriptors, and odor threshold reported in literature.

| Compound | Retention Time (min) | Concentration (mg/L) [1] | Aroma Descriptor [2] | Odor Threshold (ppb) [2] |
|---|---|---|---|---|
| *Acids* | | | | |
| Hexanoic acid | 42.0 | 25.4 | Sour, Fatty, Sweat, Cheese | 3000 |
| Octanoic acid | 46.5 | 5.1 | Rancid, Harsh, Sweaty | 3000 |
| *Alcohols* | | | | |
| 2-methylpropanol | 11.2 | 17.8 | Chemical | 250 |
| 1-Butanol | 15.6 | 548.8 | Sweet, Malty, Solventlike | 500 |
| 2-Methyl-1-butanol | 19.3 | 488.8 | Highly diluted–pleasant, Malty | 300 |
| 1-Pentanol | 22.1 | 30.5 | Fusel, Sweet, Fruity | 4000 |
| 1-Hexanol | 27.8 | 1426.5 | Herbaceous, Green, Fruity, Slightly fatty odor | 500 |
| *Cis*-3-hexenol | 28.8 | 17.9 | Grass | 70 |
| *Trans*-2-hexenol | 29.7 | 280.4 | Green, Leaf, Walnut | 400 |
| *Aldehydes* | | | | |
| Hexanal | 10.1 | 26.5 | Green, Grass | 11 |
| *Trans*-2-hexenal | 17.8 | 111.6 | Green, Apple | 17 |
| Benzaldehyde | 33.3 | 5.3 | Bitter almond, Green almond, Burnt sugar | 350 |
| *Esters* | | | | |
| Ethyl-2-methylbutanoate | 8.8 | 13.6 | Sweet, Fruity, Strawberry, Blackberry, Green apple | 0.006 |
| Butyl acetate | 9.7 | 59.4 | Red apple aroma | 66 |
| 2-Methylbutyl acetate | 12.0 | 21.4 | Fruity | 11 |
| Hexyl acetate | 20.2 | 20.7 | Herbaceous, Fruity | 2 |
| *Phenols* | | | | |
| Estragole | 37.3 | 1.8 | Licorice, Anise | 16 |

[1] Concentration expressed as pentanoic acid equivalents. [2] Odor descriptor and threshold in water, reported in the literature [13–17].

The concentration of each volatile compound was divided by the corresponding odor threshold reported in the literature, giving the Aroma Index [25], or also called odor activity value [26]. As compounds present in concentrations higher than their odor threshold are considered as aroma-contributing substances, the compounds that exhibit an Aroma Index > 1 were considered to contribute individually to apple aroma [26]. An Aroma Index higher

than 1000 corresponds to compounds that have a higher impact in aroma, presenting an odor intensity in the suprathreshold area. The odor intensity is the second dimension of the sensory perception of odorants and refers to the perceived strength or magnitude of the odor sensation. Intensity increases as a function of concentration. The relation between perceived intensity and the logarithm of odor concentration is linear. The relationship between perceived intensity and stimulus concentration was described by Fechner as a derived logarithmic function, which is the product of the Aroma Index and Weber–Fechner coefficient (2.33), plus 0.5 [27,28]. There is an intensity scale where 0 corresponds to "no odor", 1 is "very faint odor", 2 is "faint odor", 3 is "distinct odor", 4 is "strong odor", 5 is "very strong odor", and 6 or more correspond to "overwhelming odor", where the saturation is attained at 10 [26,28].

As observed in Figure 1, each compound identified in industrial apple aroma has different magnitudes of intensity. Using the Relative Aroma Index and the HS–SPME/GC–qMS peak area data as an approach to this, there were six compounds (nonanoic acid, ethanol, benzyl acetate, 2-phenylethyl acetate, *trans*-anethole, and geranylacetone) above 1 (Table 1, Figure 1a). This allows inference that these compounds were in the subthreshold area, being irrelevant to the aroma [26]. These compounds were not identified in the industrial apple aroma analyzed by SDE/GC–qMS (Table 2, Figure 1b). The Aroma Index calculated by the concentrations estimated by SDE/GC–qMS for all compounds identified showed values higher than 1. Due to the higher detection limits of the SDE technique, the concentrations determined were in the range of mg/L, showing only the major compounds present in industrial apple aroma (Table 2). The Relative Aroma Index could be an approach to complement the contribution of a wide range of volatile compounds present in the scent of apple aroma. Comparing both values (Aroma Index and Relative Aroma Index) to the same compounds identified, it was possible to validate the semi-quantification obtained by SPME. For the family of compounds present in apple aroma, the response factors and affinity of the SPME fiber polymeric material were similar, inferring that Aroma Index is comparable to the Relative Aroma Index. After this validation, SPME was used.

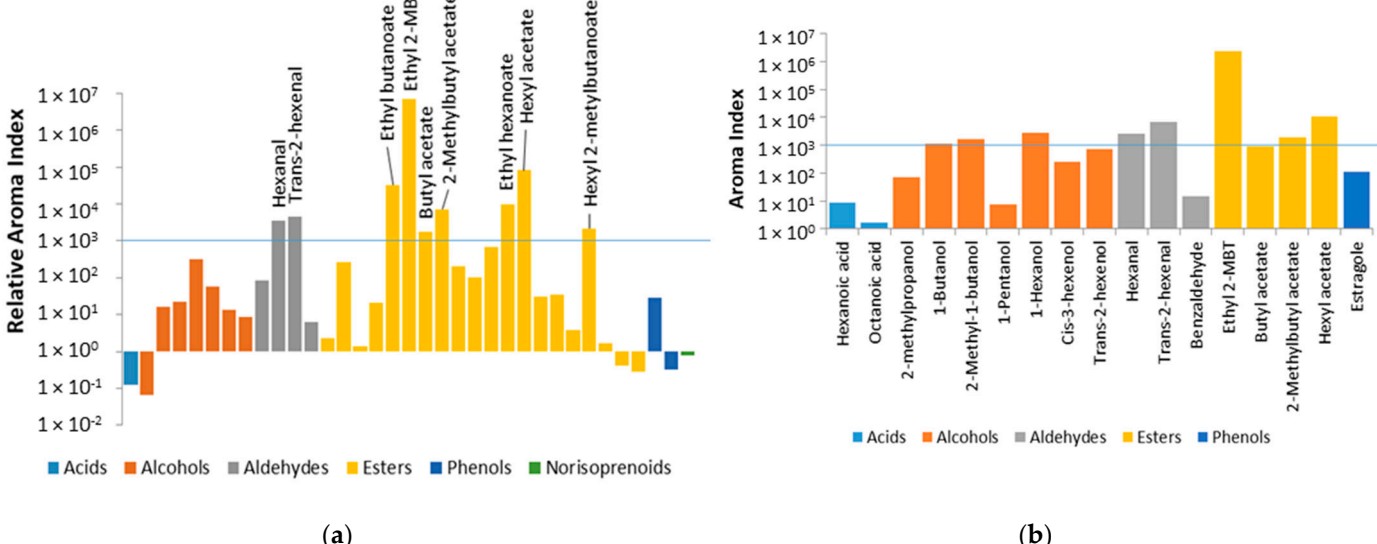

(**a**)                                                                                        (**b**)

**Figure 1.** The compounds identified in the apple aroma are plotted in a logarithmic scale against its own contribution to the aroma; values higher than 1 can contribute to aroma, values higher than 1000 present an odor intensity in the supra-threshold area (blue line). The GC peak area obtained by Headspace–Solid Phase Microextraction Gas Chromatography Combined with Quadrupole Mass Spectrometry (HS–SPME/GC–qMS) was divided by the odor threshold given by the (**a**) Relative Aroma Index (the compounds corresponding to each bar are presented in Table 1 in the same sequence); (**b**) the concentration calculated by Simultaneous Distillation Extraction (SDE)/GC–qMS divided by the odor threshold given by the Aroma Index. Ethyl-2-MBT: ethyl-2-methylbutanoate.

The compounds that most contribute to the scent of apple aroma were aldehydes and esters (Figure 1a,b), namely the ethyl-2-methylbutanoate, which presents a green apple odor descriptor with a very low odor threshold (0.006 ppb) [16]. Its Aroma Index is $2 \times 10^6$, meaning aroma intensity more than 12, within the saturation zone [26–28]. The esters, hexyl acetate and 2-methylbutyl acetate also exhibited an Aroma Index (Figure 1b) and Relative Aroma Index (Figure 1a) within the same magnitude, 4 and 3, respectively. The other four esters (ethyl butanoate, butyl acetate, ethyl hexanoate, and hexyl 2-methylbutanoate) also seem to contribute greatly to the fruity, green, and apple aroma to this concentrate apple juice coproduct. The aldehydes can also have impact in apple aroma, namely hexanal and *trans*-2-hexenal, both with a magnitude of 3, which means an aroma intensity greater than 6 [26–28].

As a coproduct of the concentrate juice industrial process, the apple aroma has a high potential of valuation due to the presence of apple aroma key compounds present in higher concentration, allowing the use of a small amount of this industrial apple aroma (e.g., 0.1% incorporation will contribute with a distinct apple odor) to aromatize different food stuffs as a natural apple aroma. This apple aroma is usually used to reconstitute the apple juice from the concentrated juice that is derived from most apple aroma compounds. In addition, apple aroma maintains the volatile profile of the single strength apple juice [3].

The apple aroma recovered from the industrial concentrate juice process has been used as food flavoring; usually there are key compounds that are used to give the price of this product. It is important to create simple and rapid methodologies to assess the quality of the aroma. In the food industry, the most common compounds used are the presence of three positive compounds: ethyl-2-methylbutanoate, hexanal, and *trans*-2-hexenal. These three compounds were described as important contributors to apple juice aroma by Ftlatch et al. [29] more than 50 years ago, and they are still used. However, the balance with some negative odor compounds are also requested to assess quality in the food industry, namely the presence of ethanol, 2-methyl-1-butanol, 3-methyl-1-butanol, isobutanol, and phenylethanol, where these compounds were classified as undesirable for apple by Dürr and Schobinger [30] almost 30 years ago. In addition, the presence of the 3-methylbuthylacetate is also considered undesirable for apple aroma, since is thought to arise from fermentation. The 3-methyl-1-butanol was found in all apple juices and aromas from different countries, within a total of 30 samples from Poland, Germany, Turkey, Romania, and China, and also 3-methylbutyl acetate was present in almost one half of the samples [3]. In our samples, only 2-methyl-1-butanol was present; however, it is considered to be a genuine apple constituent [23]. In order to assess aroma quality, some limits are regulated in juices and food matrices, and their limits should be taken into account in the aromas used as flavorings. Limits of the presence of 3-methyl-1-butanol and 3-methylbutyl acetate in fruit juice production have been a target of regulation [3]. The 3-methyl-1-butanol is a flavoring substance previously recognized as FEMA GRAS (Flavoring Extract Manufacturers Association's "Generally Recognized As Safe") with an average usual use level of 4 ppm and an average maximum use levels of 17 ppm in non-alcoholic beverages [31]. When those compounds are not present, the apple juice industry also uses other undesirable compounds at high concentrations, such as ethyl acetate (42 ppm for usual use and 67 ppm maximum for non-alcoholic beverages [31]) as negative marker for assess apple aroma quality.

Industrial Apple Aroma Volatile Compounds Quality Assessment

For the apple aroma quality assessment, five target compounds were selected, three of them contributing positively to the apple aroma, with apple odor descriptors, namely ethyl 2-methylbutanoate, hexanal, and *trans*-2-hexenal. Volatile compounds considered as undesirable to the apple aroma, e.g., 3-methyl-1-butanol and 3-methylbutyl acetate, should have been also selected [30,32]. Because those compounds were not detected, ethyl acetate, also contributing negatively to the aroma quality when present at higher concentrations, was chosen. Another compound used to assess the aroma quality was 1-butanol, as it is present in high concentrations in apple juice aroma [3,32] and can enhance the intensity

of the overall aroma [33,34]. Some of the volatile compounds selected have been already used as shelf life markers in apple juices [24]. However, as the shelf life of apple aroma has not yet been studied, in this study the apple aroma produced in January during the juice evaporation step, from apples harvested in 2013, was analyzed after 1, 5, 9, and 10 months of storage at room temperature (Figure 2).

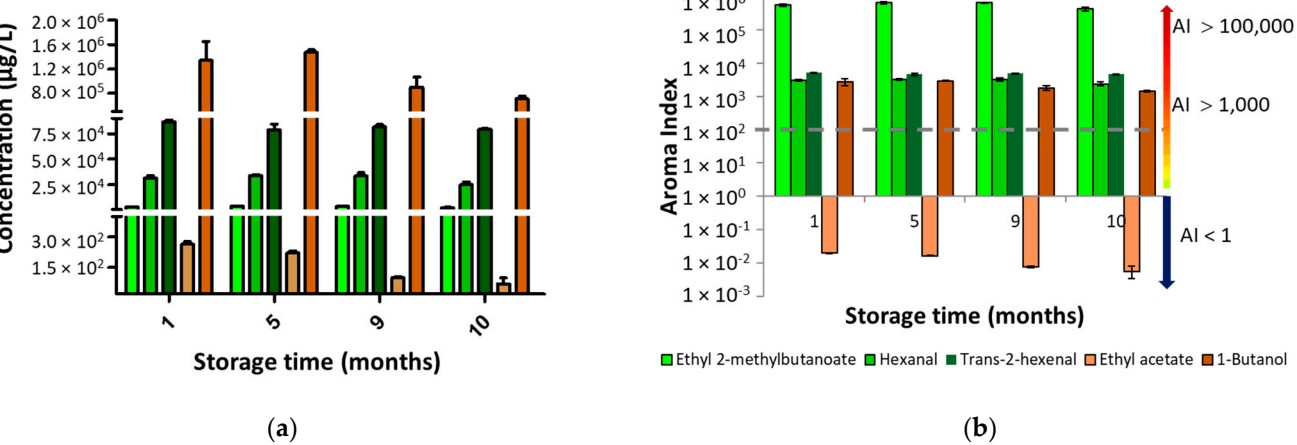

(a)                                                                                    (b)

**Figure 2.** Quality assessment of key compounds monitored during apple aroma storage at room temperature, produced in January, from apples harvested in 2013. (**a**) Concentration in µg/L of each compound; (**b**) Aroma Index (AI) calculated for each compound using the odor thresholds reported in literature. The standard deviation was represented as error bars.

The compounds that present a higher concentration in the apple aroma were 1-butanol, followed by *trans*-2-hexenal, hexanal, ethyl 2-methylbutanoate, and ethyl acetate. From the five volatile compounds monitored during 10 months, a decrease was perceived after 9 months at room temperature (Figure 2a) in only two compounds. The decrease observed was 137% and 55% in the amount of ethyl acetate and butanol, respectively. The decrease observed is related to the boiling point of the compounds, as ethyl acetate has the highest volatility and lower boiling point (68 °C). After 9 months, ethyl acetate presented less than half of its initial concentration. At 10 months of storage, the concentration decreased four and two times for ethyl acetate and butanol, respectively. 1-Butanol, with a boiling point of 117 °C, presented a decrease in a lower extent than ethyl acetate. The other three compounds, the most relevant to the apple scent aroma, almost maintained their concentration during the 10 months, all having boiling points higher than 130 °C. Ethyl 2-methylbutanoate, hexanal, and *trans*-2-hexenal at 10 months of storage decreased 38%, 28%, and 2%, respectively. Concerning the Aroma Index, those contributions were completely distinct. Ethyl 2-methylbutanoate was the compound that presented the highest Aroma Index, with a value more than 100,000 times higher than its odor threshold, showing the highest contribution to the apple aroma, with a green apple odor descriptor [16]. Hexanal, *trans*-2-hexenal, and 1-butanol, despite the different concentrations presented, have similar Aroma Index, approximately 1000 times higher than their odor thresholds (Figure 2b). Ethyl acetate presented an Aroma Index lower than 1, decreasing after 5 months from 0.1 to 0.01, which can be inferred to have a negligible influence in apple aroma, not negatively impacting apple aroma quality [32]. These results showed that apple aroma is stable at room temperature during at least 10 months.

The influence of the different raw material and technological adjustments used during the production through the year was investigated in terms of fluctuations in the apple aroma quality. The apple aroma was collected in January, produced with apples stored under controlled atmospheric conditions, and September, October, and November, using apples harvested near the fruit processing date (Figure 3). The compounds that presented the higher concentration in the apple aroma from the different productions were 1-butanol, followed by *trans*-2-hexenal; hexanal and ethyl 2-methylbutanoate presented a similar and

constant concentration over the sampling points; and ethyl acetate had the lowest concentration. Although the concentration of 1-butanol and *trans*-2-hexenal showed variation over the productions, no variation was observed for ethyl 2-methylbutanoate, the compound that is associated more with apple aroma scent (Figure 3a). Taking in consideration the Aroma Index values of the five compounds, which were very similar in all sampling points (Figure 3b), it can be inferred that the concentration variation should not be reflected in the odor perception of industrial apple aroma. A high fluctuation in concentration of the 1-butanol conferred only a slight difference in its Aroma Index.

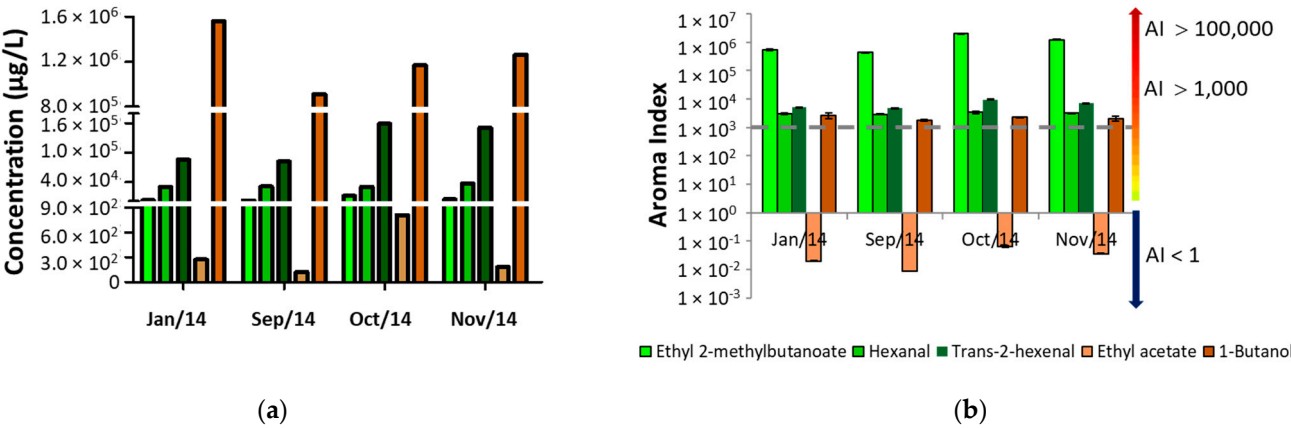

(**a**)                             (**b**)

**Figure 3.** Key compounds for quality assessment monitored in different sampling points of apple aroma resulting from juice concentrate production. The quantification was assessed by external calibration curves of each compound using HS–SPME/GC–FID methodology. (**a**) Concentration in μg/L of each compound in the different apple aroma samples; (**b**) Aroma Index calculated for each compound using the odor threshold reported in literature, assessed for the different apple aroma samples. The error bars of two replicates are represented in the figures.

### 3.2. Volatile Composition of Apple Pomace Hydrodistillate

The volatile composition of the condensed aroma compounds recovered from the wet apple pomace by hydrodistillation were compared with the industrial apple aroma, recovered from juice processing during the concentration step (Figure 4). The hydrodistillate from apple pomace contained 75 volatile compounds (Supplementary Materials, Table S1). Globally, the hydrodistillate had more intense peaks than the industrial apple aroma in compounds with lower volatility. The hydrodistillate peaks were more intense in the chemical families of acids, such as acetic, octanoic, nonanoic, and decanoic acids, associated, respectively, to the odor descriptors of vinegar, waxy, soapy, and fat [17,35]. In addition, the apple pomace hydrodistillate contained more intense peaks of alcohols, such as 1-octen-3-ol, octanol, and 6-methyl-5-hepten-1-ol, associated with the odors of fat and rubber, and aldehydes, such as 2-decenal and 2,4-heptadienal, contributing to green and fat odors [36]. When compared with the volatile profile of the industrial apple aroma, a lower intensity of high volatile esters was observed, namely those with low odor threshold (OT) and odor descriptors of apple and fruit, such as ethyl 2-methylbutanoate (OT = 0.006 μg/L) and ethyl butanoate (OT = 1 μg/L) [16].

Beyond the fact that the apple pomace hydrodistillate obtained presented a poor volatile profile quality when compared with industrial apple aroma, it presented several off-flavor compounds. These off-flavors probably arise from fermentative degradation of apple pomace due to its perishability, a consequence of the high amount of free sugars present [6,37]. The dehydration of apple pomace is a methodology able to preserve this byproduct. In addition, free sugars at high temperature are known to caramelize and provide pleasant flavors [38]. It is possible that this procedure could provide a way to valuate volatile compounds derived from apple pomace. To test this hypothesis, apple pomace was attained by conventional hot-air drying at 80 °C, with a final moisture of 4–5%

to guarantee its stabilization [39], and the volatile composition of the dried apple pomace were also assessed.

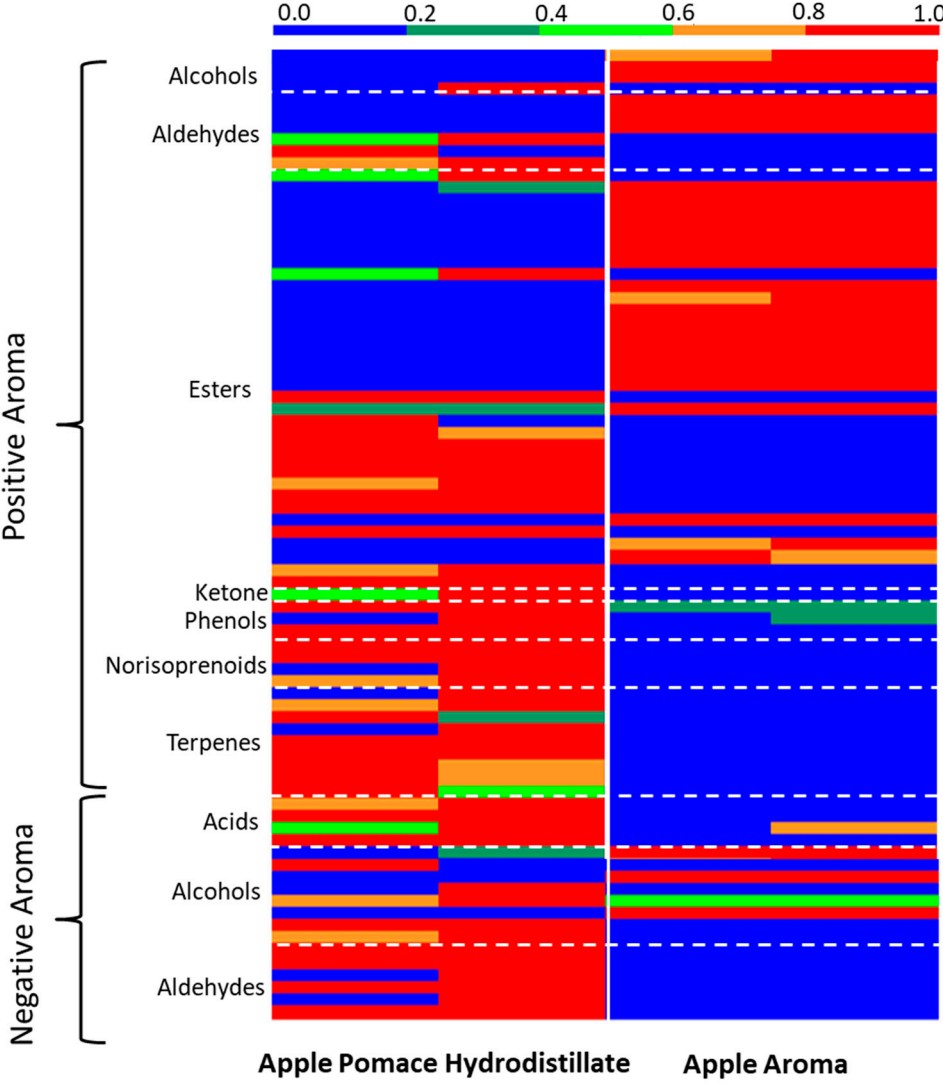

**Figure 4.** Heatmap comparing the apple hydrodistillate obtained from the wet apple pomace (from January 2014 production) and apple aroma obtained from the industrial juice processing, also from January 2014 production. Columns correspond to duplicates of each sample analyzed by HS–SPME/GC–MS. The GC peak area intensity was represented in a blue to red scale, normalized between the minimum and maximum areas for each compound. Table S1 shows the identification of the volatile compounds present.

### 3.3. Volatile Composition of Dried Apple Pomace and Its Infusions

After conventional hot-air drying at 80 °C, the apple pomace was stabilized and the volatile composition of the dried sample was analyzed by HS–SPME/GC–MS. The composition of the headspace revealed the presence of 15 compounds (Figure 5). Those present with high intensity were hexanol, 2-ethyl-1-hexanol, hexanal, furfural, benzenemethanol, benzothiazole, and 2-methoxy-6-methylpyrazine, with odor descriptors associated to green notes, toasted, caramel, and sweet [17,35,40].

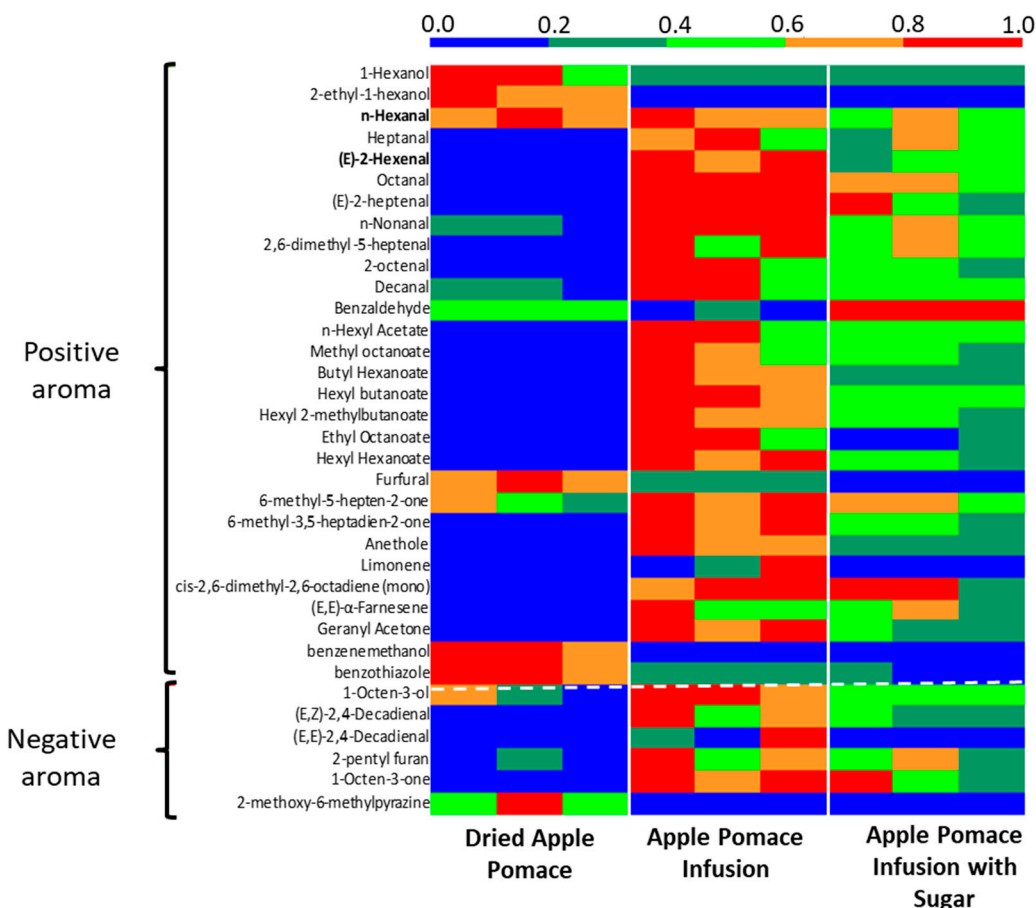

**Figure 5.** Heatmap of the volatile composition of the dried apple pomace at 80 °C and infusions obtained with dried apple pomace (from January 2014 production) without and with sugar addition. Columns correspond to triplicates of each sample analyzed by HS–SPME/GC–MS. The GC peak area intensity is represented in a blue to red scale normalized between the minimum to maximum areas for each compound.

As strategy of valorization, the dried apple pomace performance as an infusion ingredient was also evaluated. Taking in account the consumer preferences when the infusion is tasted, the volatile compounds were determined with and without sugar addition. The volatile profile of the infusions prepared with dried apple pomace allowed the identification of 35 volatile compounds in both infusions (Figure 5), a much higher number than those identified in the dried apple pomace. The infusions prepared without added sugar showed a presence of more intense peaks, namely from the chemical families of aldehydes, esters, and terpenic compounds. From these, 29 were associated with positive notes to the aroma, 20 of them with notes of fruity, apple-like, citric, and spicy. Five compounds were associated with notes of mushroom, fatty, and cucumber, usually depreciated.

The infusion with sugar addition showed a less intense volatile profile in most of the compounds present. The exception is the benzaldehyde that showed a higher intensity, possibly contributing to almond, caramel, and sweet notes to the infusion [40]. The sugar ability to modulate the aroma perception of the infusion has been reported for sucrose syrups with high °Brix [41]. The increase of sucrose concentration has been reported to decrease the flavor perception of hexanal and *trans*-2-hexenal, the green notes, while increasing the fruity notes [41]. In filtered aqueous matrices, the effect of sugar addition tends to increase the headspace partitioning of volatile compounds [42] resulting from a "salting out" effect. This relative volatility of the flavor compounds in aqueous solutions increases with their hydrophobicity [43]. In the apple infusions with sugar addition only, benzaldehyde was enriched in the headspace, which cannot be explained by its hydrophobicity [43]. It is possible that the generation of benzaldehyde during

brew preparation could arise from Strecker degradation of phenylalanine in the presence of sugars [44]. All other volatile compounds tend to decrease with the sugar addition, probably due to the presence of apple pomace matrix in the brew, a phenomenon also observed in pulp juices [45]. This phenomenon was only visible in the presence of sucrose because this sugar acts as a solvent sequestrator, which, together with the release of pectin through the cell wall matrix, increases the viscosity of the solution, preventing volatile compound diffusion to the headspace, possibly tending to be sorbed to the apple pomace hydrophobic matrix.

## 4. Conclusions

This study showed that apple industrial aroma, obtained under the concentration step of juice and recovered by evaporation/distillation, can be stored in closed containers at room temperature without noticeable loss of volatile compounds, and not requiring refrigeration. 1-Butanol, *trans*-2-hexenal, hexanal, ethyl 2-methylbutanoate, and ethyl acetate are key compounds that can be diagnostic of product quality due to their stability, maintaining the Aroma Index under the industrial process and during storage at room temperature. Four independent industrial productions from two distinct harvesting seasons showed similar Aroma Index among the key compounds, with the compound ethyl 2-methylbutanoate the highest contributor to apple scent with an Aroma Index 100,000-fold higher than its odor threshold. The coproduct, apple aroma, from the concentrate juice industry can be valuated as a high-quality flavoring agent with high potential to be a natural food ingredient.

The aroma recovered from apple pomace hydrodistillation presented a different volatile composition from the industrial apple aroma, not allowing its use due to the unpleasant aroma resulting from a high amount of long chain acids and aldehydes. Nevertheless, the aroma valuation was possible after stabilization by drying of the apple pomace. The low intense but pleasant aroma of dried apple pomace can be used for infusions preparation, providing a very pleasant and intense aroma. Sugar addition to the infusions changed the volatile profile, modulating its intensity. The apple pomace has a high potential to be valuated after its stabilization by a hot-air dried process, at 80 °C, and used as an apple infusion with a very pleasant aroma.

**Supplementary Materials:** The following are available online at https://www.mdpi.com/2076-3417/11/5/2443/s1, Table S1: GC peak area of apple hydrodistillate obtained from the wet apple pomace and apple aroma resulting from industrial juice processing, analyzed by HS–SPME/GC–MS, and divided into positive and negative aroma and chemical families according to the heatmap of Figure 4.

**Author Contributions:** Conceptualization, M.A.C. and E.C.; methodology, M.A.C., S.M.R., C.N. and E.C.; validation, M.A.C., S.M.R., C.N., and E.C.; formal analysis, M.P., M.C., and R.B.; investigation, M.P., M.C., R.B., M.A.C., and E.C.; resources, M.A.C. and S.M.R.; data curation, E.C., M.P., M.C., and R.B.; writing—original draft preparation, E.C.; writing—review and editing, M.A.C., S.M.R., C.N., R.B., M.C., and E.C.; visualization, E.C.; supervision, M.A.C. and E.C.; project administration, M.A.C., E.C., S.M.R., and C.N.; funding acquisition, M.A.C. and E.C. All authors have read and agreed to the published version of the manuscript.

**Funding:** This research was funded by Project *ProfitApple* 38162, QREN I&DT in copromotion 2013, FEDER, COMPETE. Thanks are due to the University of Aveiro and FCT—Fundação para a Ciência e a Tecnologia/MCT (Portugal) for the financial support for the QOPNA research unit [FCT UID/QUI/00062/2019] and LAQV-REQUIMTE [UIDB/50006/2020] through national funds and, where applicable, co-financed by the FEDER—Fundo Europeu de Desenvolvimento Regional, within the PT2020 Partnership Agreement, and the individual grants of Elisabete Coelho (SFRH/BPD/70589/2010) and Cláudia Nunes (SFRH/BPD/100627/2014). Elisabete Coelho (CDL-CTTRI-88-ARH/2018—REF. 049-88-ARH/2018) and Cláudia Nunes (CDL-CTTRI-88- ARH/2018-REF. 060-88-ARH/2018) also thank the research contract funded by national funds (OE), through FCT, in the scope of the framework contract foreseen in the numbers 4, 5, and 6 of the article 23, of the Decree-Law 57/2016, of 29 August, changed by Law 57/2017, of 19 July.

**Acknowledgments:** Authors also thank Oswaldo Trabulo from Indumape, S.A., for apple aroma products and byproducts supply, and Ana Maria Costa for statistical analysis.

**Conflicts of Interest:** The authors declare no conflict of interest.

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
