# Peer review of "Concentrate Apple Juice Industry: Aroma and Pomace Valuation as Food Ingredients"

_applsci, doi:10.3390/app11052443_

Round 1

Reviewer 1 Report

The manuscript titled "Concentrate apple juice industry: aroma and pomace valuation as food ingredients" describes the characterization and storage stability of juice concentrate.

The manuscript required corrections. I hope that the following suggestions and comments will be helpful.

The general remark does not show the samples from different season productions as it is described in line 84. There is info in lines 100-101 that different samples were used but there is not showed the differences between the samples. The storage temperature (4C or room temp.) is presented in the manuscript, only later is mentioned about different samples.

Lines 99-103 – the description of the concentrate production should be added – how was produced (parameters of the evaporation, time, etc).

Line 103 – how long was stored?

Line 106 – please add the condition of defrosting

 Line 108 – how long was stored?

Methods – there is no information about the repetitions et all – it is required to add this information

Lines 116, 119 compared with 167-168 – why different parameters of preparation for volatile compounds were used?

Lines 166, 169 – why the different mass of the samples was used.

Line 173 – why 2 g sugar was added?

Line 247  how was calculated Aroma Index and Relative aroma index?

Figure 1 – lack of the SD

Figure 2 – why here was chosen those samples from January and not others? Furthermore, there is a lack of statistics and SD (compare changes from a statistical point of view during storage time)

Figure 3 –there is a lack of statistics and SD (compare changes from a statistical point of view between the samples)

In my opinion, there should be described in a different order the samples – because firstly is describe Jan/14 samples during storage, without explaining why these samples were chosen. Then authors describe other samples.

To the later part - apple pomace was from which samples? Jan/14 or other? Or all mixed together?

In my opinion, the results from fig. 5 should be compared with the sensory test made by assessors.

Author Response

Answers to the Reviewer comments

Manuscript ID:  applsci-1129012

Concentrate apple juice industry: aroma and pomace valuation as food ingredients

Elisabete Coelho, Mariana Pinto, Rita Bastos, Marco Cruz, Cláudia Nunes, Sílvia M. Rocha and Manuel A. Coimbra

-Reviewer #1

Comment 1:

“The manuscript titled "Concentrate apple juice industry: aroma and pomace valuation as food ingredients" describes the characterization and storage stability of juice concentrate.

The manuscript required corrections. I hope that the following suggestions and comments will be helpful.

The general remark does not show the samples from different season productions as it is described in line 84.”

Authors answer 1:

    The authors thank the reviewer for their careful reading of the paper and constructive suggestions which help to clarify the manuscript. The season production evaluation was also included in the aims of the paper.

Comment 2:

“There is info in lines 100-101 that different samples were used but there is not showed the differences between the samples.

Authors answer 2:

According to the concentrate apple juice company the samples used over the seasons are composed from a mixture of different industrial apples, mainly composed of Royal Gala apple variety. This information was also included in the text.

Comment 3:

“The storage temperature (4C or room temp.) is presented in the manuscript, only later is mentioned about different samples.”

Authors answer 3:

Chronologically, the apple aroma produced in January was received first and analysed, evaluating the storage temperature conditions. After the definition of this parameter, the other samples were collected in the next campaign which started in September 14. The other samples allowed to evaluate the quality assessment of the apple aroma produced during juice evaporation step from different season production. This information was also included in the text.

Comment 4:

“Lines 99-103 – the description of the concentrate production should be added – how was produced (parameters of the evaporation, time, etc).”

Authors answer 4:

The available technological parameters were added to the text.

Comment 5:

“Line 103 – how long was stored?”

Authors answer 5:

The aroma samples were stored for 10 months, that was the time that volatiles were monitored in the samples. This information was added to manuscript.

Comment 6:

“Line 106 – please add the condition of defrosting”

Authors answer 6:

The conditions of defrosting were added accordingly.

Comment 7:

“ Line 108 – how long was stored?"

Authors answer 7:

The information was added in the manuscript as suggested.

Comment 8:

“Methods – there is no information about the repetitions et all – it is required to add this information”

Authors answer 8:

The number of replicates were introduced in the material and methods section.

Comment 9:

“Lines 116, 119 compared with 167-168 – why different parameters of preparation for volatile compounds were used?”

Authors answer 9:

We used two different methodologies to analyse the volatile compounds profile, the SPME and the SDE. The SPME was used in a semi-quantitative approach to assess the volatile profile of the samples. Taking in account the different response factors of the volatile compounds towards selectivity of the SPME coating fiber, an internal standard was not used. The SDE was used in a quantitative way, using an internal standard. The rational was to achieve the estimation of the concentration enabling the calculation of the Aroma Index. The concentration estimated with SDE was divided by the odour threshold of each compound obtaining the Aroma Index. The Aroma Index was then related with the Relative Aroma Index (that uses the GC peak area obtained from SPME divided by the odour threshold of each compound). Comparing both values (Aroma Index and Relative Aroma Index) to the same compounds identified, it was possible to validate the semi-quantification obtained by SPME. For the family of compounds present in apple aroma, the response factors and affinity of the SPME fiber polymeric material were similar, inferring that Aroma Index are comparable to the Relative Aroma Index. After this validation, the SPME was used to compare the apple pomace hydrodistillate and the industrial aroma. This information was also included in the manuscript.

Comment 10:

“Lines 166, 169 – why the different mass of the samples was used.”

Authors answer 10:

The different amount used of apple pomace was related to maintain the 1/β ratio in both experiments, in dried apple pomace and its infusions. The 1/β ratio (ratio of the volume of the liquid/solid phase to the headspace volume) is an important variable in SPME analysis that should be maintained the same in both experiments. Headspace volume is a critical factor determining the results in three-phase systems—liquid/solid sample-headspace-fibre coating (T. Górecki, J. Pawliszyn Analyst, 122 (1997), p. 1079). This information was added to the manuscript.

Comment 11:

“Line 173 – why 2 g sugar was added?”

Authors answer 11:

The amount of sugar was decided taking in account the sugar used to a cup of tea (aprox. 240 mL). Usually, the amount of sugar to sweeten a cup of tea is 2 packets of sugar, which each one contains 6 g of sucrose. The amount of infusion analysed was 40 mL, which correspond to the same sweeten ratio of 5% of sugar, using the 2 g of sucrose. This presuppose was also included in the manuscript.

Comment 12:

“Line 247  how was calculated Aroma Index and Relative aroma index?”

Authors answer 12:

The concentration estimated (in µg/L) with SDE was divided by the odour threshold (in µg/L) of each compound obtaining the Aroma Index. The GC peak area × 10-7 obtained from SPME was divided by the odour threshold (mg/L) of each compound obtaining the Relative Aroma Index. This information was already provided in the material and methods section of the manuscript.

Comment 13:

“Figure 1 – lack of the SD”

Authors answer 13:

The figure 1 intends to relate the Relative Aroma Index with Aroma Index, where both indices were only compared in terms of orders of magnitude, using a logarithmic scale. Small differences (in the same magnitude) are not perceived in term of a logarithmic scale.

The aroma index (using the concentration estimated with SDE) was related with the relative aroma index (that uses the GC peak area obtained from SPME). Comparing both values (Aroma Index and Relative Aroma Index) to the same compounds identified with the different methodologies was possible to validate the semi-quantification obtained by SPME. For the family of compounds present in apple aroma, the response factors and affinity of the SPME fiber polymeric material were similar, inferring that aroma index are comparable to the relative aroma index. After this validation, the SPME was used in the next analysis, e.g. to compare the apple pomace hydrodistillate and industrial aroma, because it allows to identify a higher number of compounds with a lesser detection limit than SDE.

Comment 14:

“Figure 2 – why here was chosen those samples from January and not others? Furthermore, there is a lack of statistics and SD (compare changes from a statistical point of view during storage time)”

Authors answer 14:

As samples from January were the first samples received to assess the aroma quality, they were used to evaluate the storage conditions, such as temperature. The other samples were also monitored along storage time, however the January samples were monitored during a long period, because was the first sample received. The samples received in September was monitored in only in 3 sampling points (0, 1, and 2 months of storage), and October samples were monitored only over one month of storage, with 2 sampling points. These results, not shown in the manuscript, corroborate with the more complete data set from January samples. During storage the differences in volatile composition was only perceived, in terms of concentration, after 9 months for ethyl acetate and butanol. However, the Aroma Index did not change its order of magnitude for these compounds. The objective of the present study was the quality assessment of the industrial apple aroma taking in account the impact in the aroma measured by the Aroma Index, which is irrelevant the changes in a statistical point of view if we are assuming that values with the same order of magnitude have the same contribution to the aroma. As explained in the manuscript “An Aroma Index higher than 1000 corresponds to compounds that have a higher impact in aroma, presenting an odour intensity in the supra-threshold area. The odour intensity is the second dimension of the sensory perception of odorants and refers to the perceived strength or magnitude of the odour sensation. Intensity increases as a function of concentration. The relation between perceived intensity and the logarithm of odour concentration is linear. The relationship between perceived intensity and stimulus concentration was described as a derived logarithmic function by Fechner, which is the product of the Aroma Index by Weber-Fechner coefficient (2.33) plus 0.5 [26, 27]. There is an intensity scale where 0 corresponds to “no odour”, 1 is “very faint odour”, 2 is “faint odour”, 3 is “distinct odour”, 4 is “strong odour”, 5 is ” very strong odour”, and 6 or more which correspond to “overwhelming odour”, where the saturation is attained at 10 [25, 27].” As shown in figure 2 each compound maintained each intensity scale.

The SD was represented as error bars in the figure 2, this information was added in the legend.

Comment 15:

“Figure 3 –there is a lack of statistics and SD (compare changes from a statistical point of view between the samples)”

Authors answer 15:

The objective was the quality assessment of the industrial apple aroma from different seasons taking in account the impact in the aroma measured by the Aroma Index. It is irrelevant the changes in a statistical point of view if we are assuming that values with the same order of magnitude have the same contribution to the aroma. The SD was represented as error bars in the figure 3, this information was added in the legend.

Comment 16:

“In my opinion, there should be described in a different order the samples – because firstly is describe Jan/14 samples during storage, without explaining why these samples were chosen. Then authors describe other samples.”

Authors answer 16:

Chronologically, the apple aroma produced in January was received first and analysed, evaluating the storage conditions (temperature and time). After the definition of these parameters, the other samples were collected in the next campaign which started in September. The other samples allowed to evaluate the quality assessment of the apple aroma produced during juice evaporation step from different season productions. The information of the Jan/14 samples evaluation during storage was added to the manuscript. The use of the other samples was already explained in the manuscript: “The influence of the different raw material and technological adjustments used along the production along the year was investigated in terms of fluctuations in the apple aroma quality. The apple aroma was collected in January, produced with apples stored under controlled atmospheric conditions, and September, October, and November, using apples harvested near to the fruit processing date (Figure 3).”

Comment 17:

“To the later part - apple pomace was from which samples? Jan/14 or other? Or all mixed together?”

Authors answer 17:

The apple pomace, for a correct comparison, used was from Jan/14 samples. This information was already mentioned in the material and methods section, however this information was also included in the legend of the figures 4 and 5.

Comment 18:

“In my opinion, the results from fig. 5 should be compared with the sensory test made by assessors.”

Authors answer 18:

The sensory test was only performed with consumers, during the workshop of the ProfitApple project (50 participants). During the coffee break we offer a cup of apple infusion to the participants. The infusion was tasted according to the consumer preference (with or without sugar) all participants evaluate the infusion as tasty, the only question addressed was the brew acceptance or not, as an apple infusion. This information was not added to the manuscript because was only qualitative. However, in the conclusion we use this qualitative evaluation in the sentence: “The low intense but pleasant aroma of dried apple pomace can be used for infusions preparation, providing a very pleasant and intense aroma.”

Reviewer 2 Report

The article titled: Concentrated Apple Juice: Taste and Ingredient Evaluation Marc 2 is well written and the results are fairly described and discussed.
I have a few questions / suggestions:

1. Have you checked the differences between the harvest date? In the methodology, you mentioned 4 different months, 3 of which were stored or pressed immediately. (January 2014 (controlled moshere apples -100), September 2014, October 2014 and November 2014 (apples processed after harvest or chilled) - in this article I only saw a comparison of storage temperatures - not harvest time. Future testing or not? Did you see the differences? in apple flavors from different months?

2. P9. L301 - Ftlatch et al (1967) [28] - something is wrong with the quote, the year should not be here. Same few lines below. If you want to publish this year, I suggest you do so after the word 50 years.

Author Response

Answers to the Reviewer comments

Manuscript ID:  applsci-1129012

Concentrate apple juice industry: aroma and pomace valuation as food ingredients

Elisabete Coelho, Mariana Pinto, Rita Bastos, Marco Cruz, Cláudia Nunes, Sílvia M. Rocha and Manuel A. Coimbra

Reviewer #2

Comment 1:

“The article titled: Concentrated Apple Juice: Taste and Ingredient Evaluation Marc 2 is well written and the results are fairly described and discussed.
I have a few questions / suggestions:”

Authors answer 1:

    The authors thank the reviewer suggestions and appreciation of the paper.  

Comment 2:

“1. Have you checked the differences between the harvest date? In the methodology, you mentioned 4 different months, 3 of which were stored or pressed immediately. (January 2014 (controlled moshere apples -100), September 2014, October 2014 and November 2014 (apples processed after harvest or chilled)”

Authors answer 2:

    The harvested date is not available in industrial apple samples, they are a mixture of different varieties and probably collected in different orchards and in different days. However, the samples used in January/2014 was harvested between September-October/2013. The samples used to produce concentrate apple juice and apple aroma in September 2014, October 2014, and November 2014 were processed near to the harvest time or chilled for a small period of time. This information was included in the manuscript.

Comment 3:

“- in this article I only saw a comparison of storage temperatures - not harvest time. Future testing or not? Did you see the differences? in apple flavors from different months?”

Authors answer 3:

    The harvest time was not compared. However, the processing time was, as we compared the industrial aroma produced in different seasons, corresponding to different harvesting times. The apple flavours that were monitored in the different months were the five key compounds important to define the quality of industrial apple aroma. For these 5 key compounds, the differences from different seasons were reported in the manuscript in section 3.1.1.

Comment 4:

“2. P9. L301 - Ftlatch et al (1967) [28] - something is wrong with the quote, the year should not be here. Same few lines below. If you want to publish this year, I suggest you do so after the word 50 years.”

Authors answer 4:

    The year of the publication was removed from the citations as suggested.

Reviewer 3 Report

The reviewer would like to congratulate the authors on a well documented study.

Author Response

Answers to the Reviewer comments

Manuscript ID:  applsci-1129012

Concentrate apple juice industry: aroma and pomace valuation as food ingredients

Elisabete Coelho, Mariana Pinto, Rita Bastos, Marco Cruz, Cláudia Nunes, Sílvia M. Rocha and Manuel A. Coimbra

Reviewer #3

Comment 1:

“The reviewer would like to congratulate the authors on a well documented study.”

Authors answer 1:

    The authors thank the reviewer appreciation of our work.  

Round 2

Reviewer 1 Report

The authors improve their manuscript in most places. However, there is misleading information that "Five key compounds were selected and monitored over 4 productions along one year and 10 months of storage". Only one sample from January was evaluated in storage for 10 months. In my opinion, the information about the evaluation of 4 productions should be deleted. Authors can inform about "Five key compounds were selected and monitored for 10 months of storage" and add information that the apple aroma index of concentrate from Jannuary was compared with the concentrated produced from apples collected from different harvesting periods.

This should be taken into account in the whole manuscript.

Author Response

Answers to the Round 2 Reviewer comments

Manuscript ID:  applsci-1129012

 Concentrate apple juice industry: aroma and pomace valuation as food ingredients

Elisabete Coelho, Mariana Pinto, Rita Bastos, Marco Cruz, Cláudia Nunes, Sílvia M. Rocha and Manuel A. Coimbra

-Reviewer #1

 Comment 1:

“The authors improve their manuscript in most places. However, there is misleading information that "Five key compounds were selected and monitored over 4 productions along one year and 10 months of storage". Only one sample from January was evaluated in storage for 10 months. In my opinion, the information about the evaluation of 4 productions should be deleted. Authors can inform about "Five key compounds were selected and monitored for 10 months of storage" and add information that the apple aroma index of concentrate from Jannuary was compared with the concentrated produced from apples collected from different harvesting periods.

This should be taken into account in the whole manuscript.”

Authors answer 1:

    The authors thank the reviewer comment and suggestions to improve the manuscript. The season production evaluation was revised in the manuscript were the misleading information occurred, as suggested.

Lines 16-18. “Five key compounds were selected and monitored for 10 months of storage, and also compared with other 3 productions of another season allowing to observe the same Aroma Index.”

Lines 82-85. “In the present study, the apple aroma obtained during the evaporation step from the concentrate juice process was characterized and its storage stability was evaluated. The Aroma Index of industrial apple aroma using samples from different season productions was compared.”
